# Everolimus Reduces Cancer Incidence and Improves Patient and Graft Survival Rates after Kidney Transplantation: A Multi-Center Study

**DOI:** 10.3390/jcm11010249

**Published:** 2022-01-04

**Authors:** Ryoichi Imamura, Ryo Tanaka, Ayumu Taniguchi, Shigeaki Nakazawa, Taigo Kato, Kazuaki Yamanaka, Tomoko Namba-Hamano, Yoichi Kakuta, Toyofumi Abe, Koichi Tsutahara, Tetsuya Takao, Hidefumi Kishikawa, Norio Nonomura

**Affiliations:** 1Department of Urology, Graduate School of Medicine, Osaka University, Osaka 565-0871, Japan; ryotnk0302@gmail.com (R.T.); taniguchi@uro.med.osaka-u.ac.jp (A.T.); nakazawa@uro.med.osaka-u.ac.jp (S.N.); kato@uro.med.osaka-u.ac.jp (T.K.); yamanaka@uro.med.osaka-u.ac.jp (K.Y.); abe@uro.med.osaka-u.ac.jp (T.A.); nono@uro.med.osaka-u.ac.jp (N.N.); 2Department of Nephrology, Graduate School of Medicine, Osaka University, Osaka 565-0871, Japan; namba@kid.med.osaka-u.ac.jp; 3Osaka General Medical Center, Department of Urology, Osaka 558-8558, Japan; ykakuta17@gmail.com (Y.K.); k.tsuta@gmail.com (K.T.); tetsuyatakao@gmail.com (T.T.); 4Department of Urology, Hyogo Prefectural Nishinomiya Hospital, Nishinomiya 662-0918, Japan; hidefumi69@hotmail.com

**Keywords:** cumulative incidence, everolimus, de novo cancer, kidney transplantation, mammalian target of rapamycin, survival

## Abstract

Kidney transplantation can prevent renal failure and associated complications in patients with end-stage renal disease. Despite the good quality of life, de novo cancers after kidney transplantation are a major complication impacting survival and there is an urgent need to establish immunosuppressive protocols to prevent de novo cancers. We conducted a multi-center retrospective study of 2002 patients who underwent kidney transplantation between 1965 and 2020 to examine patient and graft survival rates and cumulative cancer incidence in the following groups categorized based on specific induction immunosuppressive therapies: group 1, antiproliferative agents and steroids; group 2, calcineurin inhibitors (CNIs), antiproliferative agents and steroids; group 3, CNIs, mycophenolate mofetil, and steroids; and group 4, mammalian target of rapamycin inhibitors including everolimus, CNIs, mycophenolate mofetil, and steroids. The patient and graft survival rates were significantly higher in groups 3 and 4. The cumulative cancer incidence rate significantly increased with the use of more potent immunosuppressants, and the time to develop cancer was shorter. Only one patient in group 4 developed de novo cancer. Potent immunosuppressants might improve graft survival rate while inducing de novo cancer after kidney transplantation. Our data also suggest that everolimus might suppress cancer development after kidney transplantation.

## 1. Introduction

In patients with end-stage renal disease, kidney transplantation is a promising treatment to prevent renal failure and associated complications [1]. The introduction of new immunosuppressants has led to improved short-term patient and graft survival rates, with most recipients achieving a better quality of life after kidney transplantation [2]. However, long-term patient and graft survival rates remain insufficient [3]. Improvement of patient and graft survival rates requires the resolution of not only immunological but also non-immunological complications. Specifically, de novo cancer formation after kidney transplantation is a major complication associated with patients as well as graft survival. The incidence of de novo cancers is 3–10-fold higher in patients with organ transplants than in the general population [4]. One approach to reducing cancer risk is early detection by cancer screening [5] which, in addition to treatment, is important to protect both the patient and the kidney graft. We previously demonstrated that the overall survival rate was significantly lower in kidney transplant recipients not undergoing routine cancer screening compared to those undergoing cancer screening, highlighting the additional need for immunosuppressive protocols for cancer prevention [5]. In addition, there is an urgent need to establish immunosuppressive protocols to inhibit carcinogenesis.

The phosphatidylinositol 3-kinase (PI3K)/Akt/mammalian target of rapamycin (mTOR) signaling pathway is a major critical node for a wide range of normal cellular functions. Moreover, activation of the PI3K/Akt/mTOR pathway contributes to carcinogenesis [6] and plays a major role in regulating the growth of angiogenic tumors [7,8]. Conversely, phosphatase and tensin homolog negatively regulates the PI3K/Akt/mTOR pathway [9] and acts as a tumor suppressor. The mTOR inhibitors (mTORis), such as rapamycin, have been gaining increasing attention because of their anticancer effects, with recent studies showing their therapeutic role in angiogenic tumors such as renal cell carcinoma [8,10] and endocrine cancers [11]. Several clinical trials have demonstrated the anticancer effects of rapamycin and its analogs, including temsirolimus and everolimus [12].

The mTORis are also used as immunosuppressive agents to prevent allograft rejection in transplant patients [13,14]. Immunosuppressive induction therapy consisting of calcineurin inhibitors (CNIs), mycophenolate mofetil (MMF), and corticosteroids, is associated with improved patient and graft survival rates compared with previous treatment approaches. However, the subsequent increase in cancer prevalence has led to the consideration of adding mTORis to conventional triplet immunosuppression therapy to prevent cancer development while maintaining good immunosuppression [15].

In the present study, we aimed to investigate cancer trends and cancer-specific and all-cause mortality rates in kidney transplant recipients receiving different induction immunosuppression protocols. In addition, we examined the efficacy of the mTORi, everolimus, in cancer prevention.

## 2. Materials and Methods

### 2.1. Patient Characteristics

In this multi-center retrospective study, we reviewed the medical records of 2002 patients who underwent kidney transplantation in Osaka University Hospital (*n* = 933), Hyogo Prefectural Nishinomiya Hospital (*n* = 654), and Osaka General Medical Center (*n* = 415) between 1 June 1965 and 30 June 2020. All data were collected and analyzed on 30 September 2020 using the REDCap^®^ electronic registration software (Vanderbilt University, Nashville, TN, USA). We performed cancer screening including computed tomography and abdominal ultrasonography for all recipients once a year and registered the results to the software.

The patients were categorized into the following groups based on the type of induction immunosuppressive therapy: group 1, patients who received antiproliferative agents (azathioprine or mizoribine) and prednisolone; group 2, patients who received CNIs (cyclosporine A or tacrolimus), antiproliferative agents (azathioprine or mizoribine), and prednisolone; group 3, patients who received CNIs, mycophenolate mofetil (MMF), and prednisolone; and group 4, patients who received CNIs, MMF, mTORi, and prednisolone. This group targeted patients who started taking mTORis immediately after the kidney transplantation and continued for more than a year. But in some patients, mTORis were added at least three months after the kidney transplantation. Moreover, 21 patients who received CNIs, mTORis, and prednisolone for induction immunosuppressive therapy were included in group 4. The target trough levels of CNIs at one year after kidney transplantation were 80–100 ng/mL (cyclosporine A, group 2 and 3) and 4–6 ng/mL (tacrolimus, group 2–4).

The collected data included relevant information such as transplant and cancer history; dialysis duration before kidney transplantation; renal allograft conditions including rejection; and history of transplantation, transfusion, and comorbidities for the analyses of demographic characteristics. In addition, data were collected to determine the rates of patient survival, graft survival, cumulative cancer incidence, and cancer types.

Antilymphocyte globulin and the anti-CD25 antibody basiliximab were added to induction immunosuppressive therapy in patients undergoing kidney transplantation from 1993 to 2003 and from 2004 to 2019, respectively. Splenectomy or rituximab infusion was performed in patients undergoing ABO-incompatible kidney transplantation. Additionally, kidney biopsies were routinely performed at 3 and 12 months after kidney transplantation in patients with elevated serum creatinine levels. In patients with biopsy-proven rejection, methylprednisolone was administered for three days. The same approach was used in patients who could not be evaluated by kidney biopsy but were clinically diagnosed with rejection (e.g., >20% elevation of serum creatinine level). In patients with steroid-resistant rejection, gusperimus hydrochloride or anti-CD3 monoclonal antibody was used after methylprednisolone treatment until 2010. Starting in 2011, thymoglobulin was used as an alternative therapy for T cell-mediated rejection. For antibody-mediated rejection, plasma exchange, rituximab infusion, and intravenous immunoglobulin therapy were used.

The primary study outcome was all-cause mortality, and the secondary study outcomes were cancer-specific mortality, death-censored allograft survival rate, and cumulative cancer incidence rate. The study protocol was approved by the Institutional Review Board of Osaka University Hospital (approval no, 19475), Hyogo Prefectural Nishinomiya Hospital (approval no, H28-19), and Osaka General Medical Center (approval no, 28-2034). All procedures were performed in accordance with the 1975 Helsinki Declaration.

### 2.2. Statistical Analysis

The SPSS statistical software version 27 (IBM, Armonk, NY, USA) was used for all analyses. Categorical variables were presented as percentages or frequencies, and continuous variables were presented as means with standard deviation. Differences in group characteristics, graft survival, and overall survival among groups were compared. Univariate analyses were performed using the Mann–Whitney, Kruskal–Wallis, chi-square, and Fisher’s exact tests to compare continuous and categorical variables, as appropriate. The Kaplan–Meier method with the log-rank test was used to compare patient and graft survival rates. The statistical significance level was defined as a two-tailed *p*-value of <0.05.

## 3. Results

### 3.1. Cohort Characteristics

Of a total of 2002 recipients, 15 patients who received induction immunosuppressive monotherapy with steroids were excluded from the study. Therefore, 1987 patients were included in the final analysis (Figure 1). The summary of patient characteristics is presented in Table 1. The median follow-up durations were 21.6 ± 15.0, 18.8 ± 10.0, 10.7 ± 5.8, and 4.9 ± 2.2 years in groups 1, 2, 3, and 4, respectively. Although 24 recipients started mTORis as prescribed, they could not take it for more than a year and were included in group 3. The reasons for discontinuing oral administration were stomatitis (58.3%), followed by proteinuria, leg edema, and diarrhea. The number of elderly patients was particularly high in groups 3 and 4 (*p* < 0.001, between each group). Moreover, the number of transplantations from ABO-incompatible (*p* < 0.001, between each group) or unrelated donors (primarily spouses, *p* < 0.001, between each group) and the number of preemptive kidney transplantations (*p* < 0.001, between group 1, 2, and 3, and 1, 2, and 4, respectively) were statistically significantly higher in recent years.

In the overall cohort, 242 patients were diagnosed with de novo cancers, including 30 patients who developed two primary cancers and three patients who developed three primary cancers after transplantation (Table 2). The mean intervals between kidney transplantation and cancer diagnosis were 21.9 ± 9.7, 14.5 ± 7.2, and 6.9 ± 4.6 years in groups 1, 2, and 3, respectively. In group 4, there was only one patient who suffered de novo cancer, and the interval was 1.2 years. The mean period from transplantation to cancer diagnosis was significantly different among the four groups (*p* < 0.001). Moreover, the age of cancer diagnosis was significantly different among the four groups (*p* < 0.001). The patient with cancer in group 4 was diagnosed at 47 years of age, previously received kidney transplantation in 2013, and received multidrug immunosuppressive therapy with tacrolimus, MMF, and prednisolone for six years after the first transplantation. Finally, there was no significant difference in sex among the four groups.

The comparison of cancer types among the four group is presented in Table 3. Briefly, skin cancer (non-melanoma) was the most common malignant neoplasm in group 1 (*n* = 10, 23.8%) whereas post-transplant lymphoproliferative disorders (*n* = 21, 17.6% in group 2; *n* = 13, 16.3% in group 3), renal cell carcinoma (*n* = 12, 10.1% in group 2; *n* = 13, 16.3% in group 3), and breast cancer (*n* = 13, 10.9% in group 2; *n* = 11, 13.8% in group 3) were the most common malignant neoplasms in groups 2 and 3. Hepatocellular carcinoma (*n* = 5, 11.9% in group 1; *n* = 7, 5.9% in group 2) and gastric cancer (*n* = 5, 11.9% in group 1; *n* = 8, 6.7% in group 2), two frequent malignant neoplasms in groups 1 and 2, were less common in group 3 (hepatocellular carcinoma, *n* = 1, 1.3%; gastric cancer, *n* = 5, 6.3%). The number of patients with prostate cancer increased gradually, with 1 (2.4%), 4 (3.4%), and 5 (6.3%) patients in groups 1, 2, and 3, respectively. Lung cancer, one of the most common cancers in the general population, was diagnosed in only a few patients in each of the groups 1 (*n* = 2, 4.8%) and 2 (*n* = 1, 0.8%).

### 3.2. Cumulative de Novo Cancer Incidence Rates after Kidney Transplantation According to the Type of Induction Immunosuppressive Therapy

The 5-year cumulative de novo all cancer incidence rates after kidney transplantation were 0.0%, 1.0%, 5.3%, and 0.4% in groups 1, 2, 3, and 4, respectively (Figure 2a). Moreover, the 10- and 15-year cumulative de novo cancer incidence rates were 3.3% and 6.8% in group 1, 5.6% and 11.8% in group 2, and 11.5% and 19.3% in group 3, respectively. There were significant differences among the groups (*p* = 0.007, Gray’s test). This result suggested that the addition of mTORi to conventional immunosuppressive therapy may reduce the cancer incidence rate. Since the prognosis of skin cancer except melanoma is unlikely to generate a clinically relevant change in survival, the cumulative cancer incidence excluding non-melanoma skin cancers was also calculated. The 5-year survival rates were 0.0%, 1.1%, 4.7%, and 0.4% in groups 1, 2, 3, and 4, respectively (Figure 2b). The 10- and 15-year cumulative de novo cancer incidence rates were 3.3% and 6.9% in group 1, 5.5% and 11.3% in group 2, and 10.0% and 17.9% in group 3, respectively. Even if excluded the non-melanoma skin cancers, there were significant differences among the groups (*p* = 0.006, Gray’s test).

### 3.3. Overall and Cancer-Specific Survival Rates According to the Type of Induction Immunosuppressive Therapy

As shown in Figure 3a, the 5-year overall survival rates after kidney transplantation were 87.2%, 93.6%, 95.6%, and 96.8% in groups 1, 2, 3, and 4, respectively, with significant differences among the groups (*p* < 0.001, log-rank test) except for that between groups 3 and 4 (*p* = 0.252). Moreover, the 5-year cancer-specific survival rates after kidney transplantation were 100%, 99.4%, 99.1%, and 100% in groups 1, 2, 3, and 4, respectively (Figure 3b). The differences among the groups were not statistically significant (*p* = 0.832, log-rank test).

### 3.4. Graft Survival Rate According to the Type of Induction Immunosuppressive Therapy

The 5- and 10-year death-censored graft survival rates after kidney transplantation were 69.4% and 60.3% in group 1, 84.2% and 73.0% in group 2, and 92.8% and 85.9% in group 3, respectively (Figure 4). The 5-year graft survival rate was 97.5% in group 4, and there were no patients who had been followed up for more than 10 years in this group. The graft survival rates were significantly different among the groups (*p* < 0.001, log-rank test).

## 4. Discussion

In the present multi-center, retrospective study of 1987 patients who underwent kidney transplantation during a long time period between 1965 and 2020, we found that the cumulative cancer incidence rate after kidney transplantation increased with the introduction of new and more effective immunosuppressive drugs. These results lend further support to our previous report revealing that the aggressiveness of immunosuppressive treatment regimens or the use of potent immunosuppressives might increase the risk of de novo cancer after kidney transplantation despite the improvement of graft survival rates [16].

De novo cancer after kidney transplantation is one of the most threatening complications that reduces graft survival [16] and is a leading cause of kidney transplantation-associated mortality [17,18]. It is therefore critical to establish safe and effective immunosuppressive therapies that prevent allograft rejection as well as cancer development in patients undergoing kidney transplantation. Importantly, kidney transplant recipients who develop cancer before graft loss, i.e., those with a functioning graft at the time of cancer diagnosis, are at more than 9-fold risk of death compared to those without cancer and over 50% of recipients with cancer lose their grafts within five years following cancer diagnosis [18]. Therefore, prevention and early treatment of cancer are essential. However, discontinuation of potential cancer-promoting immunosuppressive drugs increases the risk of allograft rejection, posing a significant clinical dilemma. To address this concern, we, the authors, have started using the mTORi everolimus for induction and maintenance therapies, in addition to conventional therapies. mTORis are commonly prescribed for maintenance immunosuppression after kidney transplantation [19], and the combined use of CNIs and mTORi was reported to prevent nonmelanoma skin cancers [20,21,22,23]. Basu et al. [24] indicated that mTORis could prevent the rapid progression of post-transplantation renal cancer through the downregulation of the angiogenic cytokine vascular endothelial growth factor and the chemokine receptor CXCR3 and its ligands. mTORis act by forming a complex with the FK506-binding protein 12, and the complex inhibits mTOR, a serine-threonine kinase in the PI3K pathway, to eventually inhibit the antigenic and interleukin-stimulated activation and proliferation of T cells [25]. Several other mechanisms of cancer prevention by mTORis were also reported. For example, mTORis induce cell cycle arrest at the G1 checkpoint [26]. Additionally, everolimus, the mTORi included in the present study, has also been approved for use in several malignancies including advanced metastatic renal cell cancer [10,27]; gastric, intestinal, and pancreatic neuroendocrine tumors [11]; and subependymal giant cell astrocytoma [28].

A study previously reported that patients on mTORi-based therapy had a 59% reduced relative risk of developing new cancers (relative risk, 0.412; 95% confidence interval, 0.256–0.663) than those on CNI-based therapy [23], suggesting the possibility of replacing CNIs with mTORis in kidney transplant recipients. However, this approach may not be appropriate. Complete avoidance or withdrawal of CNIs could be associated with a greater risk of acute or chronic rejection and graft failure [29,30]. Conversion from mTORis to CNIs has been reported to be associated with increased risk of donor-specific antibody production and reduced allograft survival rate because of chronic antibody-mediated rejection [31]. Therefore, we recognize that conversion from CNIs to mTORis may not be effective in improving graft survival rate. Moreover, MMF is effective in suppressing deoxyribonucleic acid production and the onset of antibody-mediated rejection, which is difficult not only in early diagnosis but also in treatment [32,33]. In the present study, we found that quadruple induction immunosuppressive therapy, including the three drugs used in group 3 plus everolimus, was associated with not only a reduction in cancer incidence but also with an improvement in recipient and graft survival rates. The quadruple therapy appears to provide strong, albeit brief, immunosuppression; however, since the introduction of MMF and prednisolone, the CNI dose has been reduced (5–8 ng/mL within 3 months, 4–5 ng/mL at 4–12 months after kidney transplantation). Therefore, complications, such as infectious diseases, caused by potent immunosuppressive status did not increase (data not shown).

Of course, incidence and survival rates of specific cancers are not determined by whether mTORi is used for induction immunosuppressive therapy. The background characteristics contributing to specific cancer incidences should be carefully considered. For example, our results indicated that the incidence of hepatocellular carcinoma (HCC) was lower in group 3 than in groups 1 and 2. Hepatitis B and C virus infections are the main causes of HCC. Our data also showed that 11 of 13 cases were infected with at least hepatitis B or C. (type B only; *n* = 2, type C only; *n* = 6, both; *n* = 3, respectively). Blood transfusion associated with end-stage renal disease is also a cause of hepatitis. It is strongly likely that the decrease in the rate of hepatitis due to the development of erythropoiesis-stimulating agents and direct-acting antivirals has led to a decrease in the rate of HCC after kidney transplantation. On the other hand, the rates of renal cell carcinoma and prostate cancer increased in group 3 compared to groups 1 and 2. The duration of hemodialysis before kidney transplantation was also significantly longer in group 3. The preoperative dialysis period before kidney transplantation was previously reported to be correlated with the rate of renal cell carcinoma [34], as reflected in our findings. Conversely, the increase in the number of elderly recipients in recent years was likely linked to the increase in the incidence of prostate cancer observed in the present study. The average recipient age at the time of kidney transplantation was significantly higher in group 4 than that in other groups. Therefore, it is possible that mTORis might suppress the development of prostate cancer.

We have performed cancer screening on most patients so far, including the current cohort [5]. It is not possible to detect cancer by screening in all patients. However, we believe that cancer screening is a major underlying cause of the lack of a difference in cancer-specific survival rates among the study groups. In addition to cancer screening, we believe that suppressing the onset of de novo cancers using mTORis in combination with aggressive immunosuppressive therapy after kidney transplantation is critical. Several studies have also reported the beneficial effect of mTORis in cardiovascular diseases [35,36] and infections [37,38,39]. Cancers, cardiovascular diseases, and infections are leading causes of death after kidney transplantation [40]. Therefore, mTORis might contribute to the further improvement of survival rates by suppressing the onset of these diseases.

mTORi was previously used as a treatment for renal cell carcinoma (10 mg/day). However, various adverse events (e.g., stomatitis; 40–60%) occurred at high rates, and it was often difficult to continue [27,41,42]. Fortunately, when used as an immunosuppressant, the dose (1.5 mg/day) is about one-eighth, and the incidence of adverse events is relatively low [43]. In our study, mTORi was able to continue for a long time without interruption in most cases. However, mTORi adverse events can significantly impair the recipient’s quality of life and should be carefully considered for risks and benefits and discontinued if necessary.

The present study has several notable strengths. This study was based on a large population of kidney transplant recipients from several high-volume centers. The study analyses were based on a comprehensive dataset with a long duration and very few missing values. The completeness of the dataset suggests that selection and ascertainment biases in exposure and study factors were minimal. However, the present study has several limitations that should also be acknowledged. First, this was a retrospective study, and the observation period of group 4 was notably shorter than that of the other groups. Therefore, although mTORi has been shown to reduce cancer morbidity with short-term observations, we believe that the efficacy of mTORi needs to be reassessed at long-term observations. Second, despite the adjustment for all confounding factors, there may be unmeasured residual effects such as details on the dose and duration of immunosuppressants used for the treatment of primary disease and incomplete details on smoking habits and alcohol consumption, which might have altered the strength and magnitude of association of cancer risk after transplantation. Third, some patients in groups 1, 2, and 3 received mTORis after the cancer diagnosis, although the treatment was initiated several years after the kidney transplantation. Finally, cancer treatment approaches and reductions in immunosuppressant doses in patients who developed cancer were determined by the attending physician in each case. However, despite these limitations, the current study findings are important as they are based on the largest Japanese cancer registry data of patients with kidney transplants. Especially, it is noteworthy that in group 4, the prevalence of de novo cancers at the early phase after kidney transplantation was clearly lower than in the other groups, despite the significantly older recipients.

## 5. Conclusions

Despite the improvement of patient and graft survival rates with the increasing potency of immunosuppressants, the cumulative cancer incidence rate after kidney transplantation increased, with a tendency for a shorter time from kidney transplantation to cancer development. The addition of mTORis to conventional induction immunosuppressive therapy was not only associated with improved patient and graft survival rates but also with reduced cancer incidence, at least in the short term after kidney transplantation. The current study findings have potential clinical significance, including the optimization of effective immunosuppressive therapies to prevent the development of post-transplantation cancer, which requires further long-term studies.

## Figures and Tables

**Figure 1 jcm-11-00249-f001:**
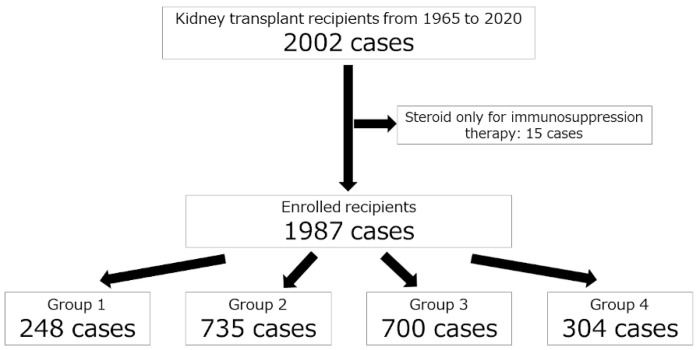
A diagram describing the study population.

**Figure 2 jcm-11-00249-f002:**
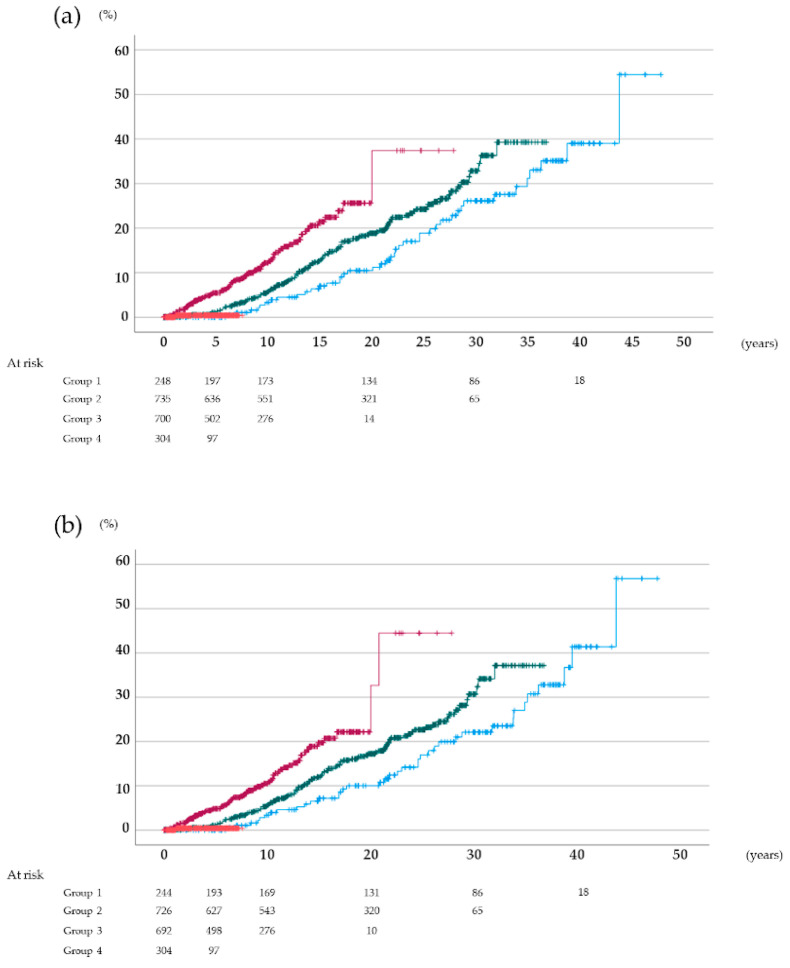
Cumulative cancer incidence rates after kidney transplantation. (**a**) all cancers, (**b**) all cancers except non-melanoma skin cancer: Blue, group 1; green, group 2; dark red, group 3; vermilion, group 4.

**Figure 3 jcm-11-00249-f003:**
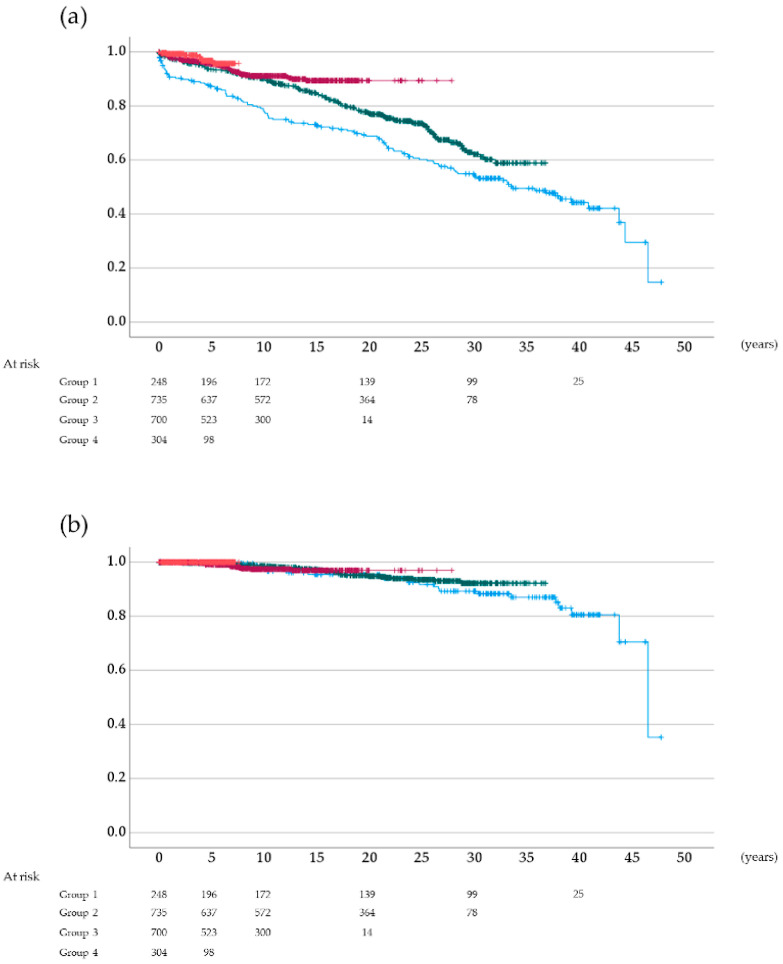
(**a**) Overall and (**b**) cancer-specific survival rates after kidney transplantation: Blue, group 1; green, group 2; dark red, group 3; vermilion, group 4.

**Figure 4 jcm-11-00249-f004:**
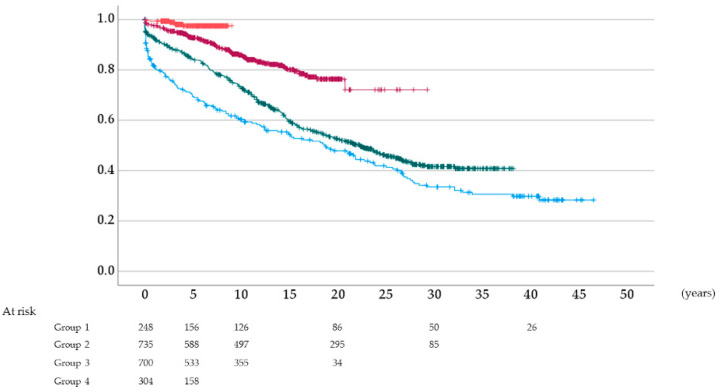
Death-censored graft survival rates after kidney transplantation: Blue, group 1; green, group 2; dark red, group 3; vermilion, group 4.

**Table 1 jcm-11-00249-t001:** Demographic characteristics of transplant recipients.

		Group 1	Group 2	Group 3	Group 4	*p*-Value
The Number of Patients		248	735	700	304	
Follow-up duration (yr)		21.6 (15.0)	18.8 (10.0)	10.7 (5.8)	4.9 (2.2)	<0.001
Recipient age (yr)	<30	141 (63.2)	248 (36.1)	108 (16.3)	25 (9.5)	<0.001
	30–45	100 (34.2)	367 (47.6)	265 (33.8)	89 (30.4)	
	46–60	7 (2.6)	113 (15.5)	234 (35.7)	128 (36.7)	
	>61	0 (0.0)	7 (0.8)	93 (14.2)	62 (23.4)	
Recipient sex	Female	89 (35.9)	294 (40.0)	268 (38.3)	113 (37.2)	0.648
	Male	159 (64.1)	441 (60.0)	432 (61.7)	191 (62.8)	
Donor type	living	214 (86.3)	539 (73.3)	615 (87.9)	273 (89.8)	<0.001
	deceased	34 (13.7)	196 (26.7)	85 (12.1)	31 (10.2)	
Blood relation	Yes	214 (86.3)	530 (72.1)	432 (61.7)	146 (48.0)	<0.001
ABO blood-type	compatible	248 (100.0)	715 (97.3)	548 (78.3)	201 (66.1)	<0.001
	incompatible	0 (0.0)	20 (2.7)	152 (21.7)	103 (33.9)	
Number of HLA mismatches	A, B, DR	0.64 (1.01)	1.39 (1.06)	1.80 (1.26)	1.79 (1.29)	<0.001
Body mass index (kg/m^2^)		21.5 (1.47)	21.3 (1.55)	21.3 (2.57)	21.4 (3.33)	0.417
Dialysis duration (yr)	0	7 (3.1)	17 (2.6)	109 (18.6)	89 (26.6)	<0.001
	<1, ≤1	91 (36.8)	202 (27.0)	153 (21.7)	59 (20.3)	
	1–3, ≤3	92 (25.0)	215 (17.1)	147 (12.3)	61 (16.5)	
	3–5, ≤5	38 (27.6)	111 (28.0)	69 (16.3)	23 (17.7)	
	5–10, ≤10	19 (7.0)	130 (17.3)	86 (12.3)	31 (7.0)	
	10–20, ≤20	1 (0.4)	51 (6.8)	94 (13.1)	29 (8.2)	
	>20	0 (0.0)	9 (1.1)	42 (5.7)	12 (3.8)	
Calcineulin inhibitors	Cyclosporine	0 (0.0)	552 (75.1)	212 (30.3)	17 (5.6)	<0.001
	Tacrolimus	0 (0.0)	183 (24.9)	488 (69.7)	287 (94.4)	
	none	248 (100.0)	0 (0.0)	0 (0.0)	0 (0.0)	
Antiproliferative agents	Azathioprine	248 (100.0)	361 (49.1)	0 (0.0)	1 (0.3)	<0.001
	Mizoribine	0 (0.0)	321 (43.7)	0 (0.0)	3 (1.0)	
	Mycophenolate mofetil	0 (0.0)	0 (0.0)	700 (100.0)	300 (98.7)	
	none	0 (0.0)	53 (7.2)	0 (0.0)	0 (0.0)	
mTOR inhibitor (everolimus)	induction therapy	0 (0.0)	0 (0.0)	0 (0.0)	155 (51.0)	<0.001
	add-on within 3 months	0 (0.0)	0 (0.0)	0 (0.0)	149 (49.0)	
History of rejection	Yes	145 (58.5)	519 (70.6)	165 (23.6)	31 (10.2)	<0.001
History of transplantation	Yes	6 (2.4)	17 (2.3)	42 (6.0)	21 (7.6)	<0.001
History of transfusion	Yes	87 (35.1)	116 (15.8)	148 (21.1)	70 (23.0)	<0.001

Categorical variables are presented as frequencies and/or percentages, and continuous variables are presented as means with standard deviation.

**Table 2 jcm-11-00249-t002:** Demographic characteristics of transplant recipients with de novo cancers after kidney transplantation.

		Group 1	Group 2	Group 3	Group 4	*p*-Value
The Number of Patients		42 (16.9)	119 (16.2)	80 (11.7)	1 (0.3)	
Number of yrs from KTP to diagnosis of cancer (yr)		21.9 (9.7)	14.5 (7.2)	6.9 (4.6)	1.2 (-)	<0.001
Recipient age (yr)	<30	20 (47.6)	23 (19.3)	10 (12.5)	0 (0.0)	<0.001
	30–45	19 (45.2)	72 (60.5)	25 (31.2)	0 (0.0)	
	46–60	3 (7.2)	23 (19.3)	32 (40.0)	1 (100.0)	
	>61	0 (0.0)	1 (0.9)	13 (16.3)	0 (0.0)	
Recipient sex	Female	15 (35.7)	52 (43.7)	38 (47.5)	0 (0.0)	0.501
	Male	27 (64.3)	67 (56.3)	42 (52.5)	1 (100.0)	
Donor type	living	36 (85.7)	81 (68.1)	72 (90.0)	1 (100.0)	0.02
	deceased	6 (14.3)	38 (31.9)	8 (10.0)	0 (0.0)	
Blood relation	Yes	36 (85.7)	81 (67.8)	48 (60.0)	1 (100.0)	0.028
ABO blood-type	compatible	42 (100.0)	117 (98.3)	59 (73.8)	1 (100.0)	<0.001
	incompatible	0 (0.0)	2 (1.7)	21 (26.2)	0 (0.0)	
Number of HLA mismatches	A, B, DR	1.17 (1.32)	2.02 (1.02)	2.99 (1.51)	3.00 (-)	<0.001
Body mass index (kg/m^2^)		21.8 (1.28)	21.3 (1.58)	21.3 (2.36)	27.64 (-)	0.11
Dialysis duration (yr)	0	2 (4.8)	3 (2.5)	6 (7.5)	1 (100.0)	0.02
	≤1	9 (21.4)	22 (18.5)	13 (16.3)	0 (0.0)	
	1<, ≤2	10 (23.8)	18 (15.1)	9 (11.3)	0 (0.0)	
	2<, ≤5	13 (31.0)	32 (26.9)	20 (25.0)	0 (0.0)	
	5<, ≤10	6 (14.3)	23 (19.3)	13 (16.3)	0 (0.0)	
	10<, ≤20	0 (0.0)	12 (10.1)	13 (16.3)	0 (0.0)	
	>20	0 (0.0)	3 (2.5)	6 (7.5)	0 (0.0)	
Calcineulin inhibitors	Cyclosporine	0 (0.0)	88 (73.9)	34 (41.4)	0 (0.0)	<0.001
	Tacrolimus	0 (0.0)	31 (26.1)	46 (58.5)	1 (100.0)	
	none	42 (100.0)	0 (0.0)	0 (0.0)	0 (0.0)	
Antiproliferative agents	Azathioprine	42 (100.0)	69 (58.0)	0 (0.0)	0 (0.0)	<0.001
	Mizoribine	0 (0.0)	44 (42.0)	0 (0.0)	0 (0.0)	
	Mcophenolatemofetil	0 (0.0)	0 (0.0)	80 (100.0)	1 (100.0)	
	none	0 (0.0)	6 (5.1)	0 (0.0)	0 (0.0)	
mTOR inhibitor	everolimus	0 (0.0)	0 (0.0)	0 (0.0)	1 (100.0)	
History of rejection	Yes	24 (57.1)	85 (71.4)	24 (30.5)	0 (0.0)	<0.001
History of transplantation	Yes	1 (2.4)	4 (3.4)	6 (8.5)	1 (100.0)	<0.001
History of transfusion	Yes	23 (54.8)	26 (21.8)	15 (23.2)	1 (100.0)	<0.001

Categorical variables are presented as frequencies and/or percentages, and continuous variables are presented as means with standard deviation.

**Table 3 jcm-11-00249-t003:** Distribution of cancer types.

	Group 1	Group 2	Group 3	Group 4
The number of recipients	248	735	700	304
Total cancer-positive recipients	42 (16.9)	119 (16.2)	80 (11.7)	1 (0.3)
Double cancer-positive recipients	6	12	9	0
Triple cancer-positive recipients	4	1	0	0
Type of cancer				
PTLD	3	21	13	1
renal cell carcinoma	2	12	13	0
breast cancer	4	13	13	0
skin cancer (melanoma)	0	0	1	0
skin cancer (non-melanoma)	10	12	9	0
prostate cancer	1	5	5	0
colorectal cancer	5	7	6	0
uterus cancer	2	10	5	0
gastric cancer	5	8	5	0
urothelial cancer	2	6	4	0
thyroid cancer	1	6	3	0
tongue cancer	3	7	2	0
pancreas cancer	0	2	2	0
hepatocellular carcinoma	5	7	1	0
lung cancer	2	1	0	0
ovarian cancer	1	1	0	0
vaginal cancer	0	1	0	0
anal cancer	0	1	0	0
others	10	13	7	0
Total	56	133	89	1

PTLD, post-transplant lymphoproliferative disorders.

## Data Availability

The data presented in this study are available upon request from the corresponding author. The data are not publicly available due to privacy restrictions.

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
