# Peer review of "Everolimus Reduces Cancer Incidence and Improves Patient and Graft Survival Rates after Kidney Transplantation: A Multi-Center Study"

_jcm, 2022, doi:10.3390/jcm11010249_

Round 1
Reviewer 1 Report
The authors combine three centers in their country and follow cancer occurrences in four groups with different maintenance immunosuppression. The authors show that those on mTORi have lower cancer incidence, though this group has the shortest follow-up by far.
Figure 1 title needs to be changed.
Figure 2 title needs to be changed. (Start with Cumulative cancer…)
Can you remove the non-melanomatous skin cancers from the graph as these are usually not reported with the other cancers due to their low malignancy potential?
There are so few patients in group 4 out more than 3 years that the graph is misleading. Is it possible to have the number of patients at risk on the graph?
There is traditionally a high discontinuation rate with patients who start on mTORi. Please show how many of your patients were able to stay on mTORi or had to change, and at what time after transplant?
Skin cancer needs to be subdivided into melanoma and non-melanoma skin cancer. When you do this, do the statistics of the group comparisons change?
Did the patients with hepatocellular cancer have Hepatitis B and/or Hepatitis C?
Induction therapy is usually considered the antibody or steroid treatment given the first week of transplant to help prevent rejection as you start maintenance therapy. You place mTORi in the group of ‘induction’ immunosuppression when I believe you mean to say that is was the initial maintenance agent used.
Author Response
Response to Reviewer 1 Comments
Here is a point-by-point response to the reviewers’ comments and concerns.
We used a red-colored font to show changes to the text in the revised manuscript.
Thanks to your comments, I am grateful that this paper has been further refined.
Point 1: Figure 1 title needs to be changed. 

Response 1: We agree the reviewer’s comment and change the title.
Point 2: Figure 2 title needs to be changed. (Start with Cumulative cancer…)
Response 2: We agree the reviewer’s comment and change the title as reviewer 1 suggested.
Point 3: Can you remove the non-melanomatous skin cancers from the graph as these are usually not reported with the other cancers due to their low malignancy potential?
Response 3: We agree the reviewer’s comment. We created two graphs in Figure1 on page 9. One was the cumulative cancer incidence of all cancers and the other was the cumulative cancer incidence of all cancers except non-melanoma skin cancers.
Point 4: There are so few patients in group 4 out more than 3 years that the graph is misleading. Is it possible to have the number of patients at risk on the graph?
Response 4: We indicated the number of patients at risk on each graph.
Point 5: There is traditionally a high discontinuation rate with patients who start on mTORi. Please show how many of your patients were able to stay on mTORi or had to change, and at what time after transplant?
Response 5: Oral administration was rarely unsustainable, as was the case with renal cell carcinoma. This is thought to be due to the low dose of mTORi after organ transplantation. Nevertheless, some recipients were unable to continue mTORi due to adverse events. In all cases, mTORi was eliminated within 1 year of the start of oral administration. (Line 133 of page 3)
Point 6: Skin cancer needs to be subdivided into melanoma and non-melanoma skin cancer. When you do this, do the statistics of the group comparisons change?
Response 6: We divided the data on Table3. Except one case, the skin cancers were non-melanoma. Therefore, the stastistics of the group comparison did not change.
Point 7: Did the patients with hepatocellular cancer have Hepatitis B and/or Hepatitis C?
Response 7: Yes. Eleven of 13 cases had. We indicated on line326 of page 12 that “Our data also showed that 11 of 13 cases were infected with at least Hepatitis B or C. (type B only; n=2, type C only; n=6, both; n=3, respectively)”.
Point 8: Induction therapy is usually considered the antibody or steroid treatment given the first week of transplant to help prevent rejection as you start maintenance therapy. You place mTORi in the group of ‘induction’ immunosuppression when I believe you mean to say that is was the initial maintenance agent used.
Response 8: Thank you for your excellent comment. In Japan, becauce of public health insurance, we are not be able to use Thymoglobulin for induction therapy. Therefore, just before or just after kidney transplantation, we start to use CNI, MMF, mTORi, steroid. For us, using these immunosuppressants is the induction therapy. I think that the content of introduction therapy depends on the system of each country. Your suggestions are very important and will be used for future reference.
Reviewer 2 Report
The authors have investigated the role of more potent immunosuppressants and everolimus in suppressing the development of cancer development after kidney transplantation. The results are interesting, demonstrating that patient and graft survival increased with the use of more potent immunosuppressants, while interval from development of cancer was shorter. Furthermore, everolimus, mTOR inhibitor, seems to suppress cancer development after KT in this study.
However, this study has major flaws in the follow-up periods.
Follow-up periods of group 4 is just 3.6 years. For the cancer to develop, at least 5 years follow-up is necessary. Group 4 consists of patients with age more than 61 years in large percentage compared with other groups. This implies that even though this group develops cancer, the progression would be very slow and the detection rate would be low. This makes perfect situation that looks as if the incidence of cancer development is low at group 4. This is also supported by the Table 2, where interval from development of cancer is shorter in group 4 compared to other groups.
The shorter the follow-up periods, the shorter time to develop cancers. This implies that the authors should compensate this time factor in order to state their statement. I believe that without this effort and merely stating that difference in follow-up periods is limitation, this manuscript has serious statistical flaws although the idea is very interesting.
Furthermore, discussion of adverse side effects by more potent immunosuppressants and mTOR inhibitor should be also discussed. mTOR inhibitor when used in RCC patients, the most common adverse reactions with incidence of more than 30% were stomatitis, infections, rash, fatigue, diarrhea, and decreased appetite. The Grade 3-4 adverse reactions are reported with incidence ≥ 2%, include stomatitis, infections, hyperglycemia, fatigue, dyspnea, pneumonitis, and diarrhea. Therefore, we should not overlook this AEs by everolimus.
Author Response
Response to Reviewer 2 Comments
Here is a point-by-point response to the reviewers’ comments and concerns.
We used a red-colored font to show changes to the text in the revised manuscript.
Thanks to your comments, I am grateful that this paper has been further refined.
Point 1: Follow-up periods of group 4 is just 3.6 years. For the cancer to develop, at least 5 years follow-up is necessary. Group 4 consists of patients with age more than 61 years in large percentage compared with other groups. This implies that even though this group develops cancer, the progression would be very slow and the detection rate would be low. This makes perfect situation that looks as if the incidence of cancer development is low at group 4. This is also supported by the Table 2, where interval from development of cancer is shorter in group 4 compared to other groups.
Response 1: We agree the reviewer’s comment of follow-up periods. In the first paper, we analysed on September 30 2020 (endpoint). We reparsed the database with November 30, 2021 as the endpoint. As a result, the average observation period of Group 4 was extended to 4.9 years. Fortunately, this result did not significantly change the results obtained in the first draft, nor does the conclusion change.
Many papers reported that risk factors for de novo cancer after kidney transplantation were recipients of the elderly (e.g., Hung YM, et al. Urology 69(6), 2007, 1041-1044). The average recipient age at kidney transplantation in Group 4 was high and was a risk factor for de novo cancer. In addition, at these three facilities, cancer screening is conducted once a year for each recipient. Therefore, I presume that your comment “This makes perfect situation that looks as if the incidence of cancer development is low at group 4” is not always correct.
One case in Group 4 was a secondary transplant case (on line 164 of page 5), which was affected by the initial immunosuppression and might be diagnosed with de novo cancer at the early period after kidney transplantation.
Point 2: The shorter the follow-up periods, the shorter time to develop cancers. This implies that the authors should compensate this time factor in order to state their statement. I believe that without this effort and merely stating that difference in follow-up periods is limitation, this manuscript has serious statistical flaws although the idea is very interesting.
Response 2: We agree the reviewer’s comment. We added a strong comment that the effectiveness of mTORi should be reassessed after long-term follow-up. (On line 365 of page 12)
Point 3: Furthermore, discussion of adverse side effects by more potent immunosuppressants and mTOR inhibitor should be also discussed. mTOR inhibitor when used in RCC patients, the most common adverse reactions with incidence of more than 30% were stomatitis, infections, rash, fatigue, diarrhea, and decreased appetite. The Grade 3-4 adverse reactions are reported with incidence ≥ 2%, include stomatitis, infections, hyperglycemia, fatigue, dyspnea, pneumonitis, and diarrhea. Therefore, we should not overlook this AEs by everolimus. 

Response 3: We agree the reviewer’s comment. The dose of everolimus for kidney transplantation is about one-eighth that when used for the treatment of renal cell carcinoma. Therefore, the incidence of adverse events when used as an immunosuppressant is relatively low. However, as reviewer pointed out, there are some cases in which mTORi should be eliminated for adverse events. We added the comments about mTORi on line 351 of page 12. In addition, we added our data about the elimination rate of mTORi on line 133 of page 3.
Round 2
Reviewer 2 Report
No further comments